# See, Hear, and Feel:
# Smart Sensory Fusion for Robotic Manipulation

**Hao Li**[1]* **Yizhi Zhang**[1]* **Junzhe Zhu**[1] **Shaoxiong Wang**[2] **Michelle A Lee**[1]
**Huazhe Xu**[1] **Edward Adelson**[2] **Li Fei-Fei**[1] **Ruohan Gao**[1]† **Jiajun Wu**[1]†
[1]Stanford University    [2]Massachusetts Institute of Technology
*Equal contribution.    †Equal advising.

**Abstract:** Humans use all of their senses to accomplish different tasks in everyday activities. In contrast, existing work on robotic manipulation mostly relies on one, or occasionally two modalities, such as vision and touch. In this work, we systematically study how visual, auditory, and tactile perception can jointly help robots to solve complex manipulation tasks. We build a robot system that can *see* with a camera, *hear* with a contact microphone, and *feel* with a vision-based tactile sensor, with all three sensory modalities fused with a self-attention model. Results on two challenging tasks, dense packing and pouring, demonstrate the necessity and power of multisensory perception for robotic manipulation: vision displays the global status of the robot but can often suffer from occlusion, audio provides immediate feedback of key moments that are even invisible, and touch offers precise local geometry for decision making. Leveraging all three modalities, our robotic system significantly outperforms prior methods.

**Keywords:** Multisensory Perception, Robotic Manipulation, Robot Learning

## 1 Introduction

Imagine you are savoring tea in a peaceful Zen garden: a robot sees your empty cup and starts pouring, hears the increase of the sound pitch as the water level rises in the cup, and feels with its fingers around the handle of the teapot to tell how much tea is left and control the pouring speed. For both humans and robots, multisensory perception with vision, audio, and touch plays a crucial role in everyday tasks: vision reliably captures the global setup, audio sends immediate alerts even for occluded events, and touch provides precise local geometry of objects that reveal their status.

Though exciting progress has been made on teaching robots to tackle various tasks [1, 2, 3, 4, 5], limited prior work has combined multiple sensory modalities for robot learning. There have been some recent attempts that use audio [6, 7, 8, 9] or touch [10, 11, 12, 13, 14] in conjunction with vision for robot perception, but no prior work has simultaneously incorporated visual, acoustic, and tactile signals—three principal sensory modalities, and study their respective roles on challenging multisensory robotic manipulation tasks.

We aim to demonstrate the benefit of fusing multiple sensory modalities for solving complex robotic manipulation tasks, and to provide an in-depth study of the characteristics of each modality and how they complement each other. Towards this end, we equip a Franka Emika Panda robot arm with a camera, a contact microphone, and a vision-based tactile sensor to receive visual, acoustic, and tactile signals, respectively. We tackle two challenging robotic manipulation tasks: 1) *dense packing*, where the goal is to insert a target object into a densely cluttered box (see Fig. 1), and 2) *pouring*, where the goal is to pour tiny beads from one cylinder container to the other and reach the target level. We choose these two tasks as our case studies due to their broad real-world applications, complexities, and dependence on multisensory perception.

We propose a new multisensory self-attention model for fusing visual, acoustic, and tactile sensory data, and use imitation learning for action prediction. Our model uses a self-attention mechanism to attend both across modalities and across time, outperforming prior fusion methods and its ablated versions without certain modalities by a large margin. Through a series of quantitative and quali-

6th Conference on Robot Learning (CoRL 2022), Auckland, New Zealand.

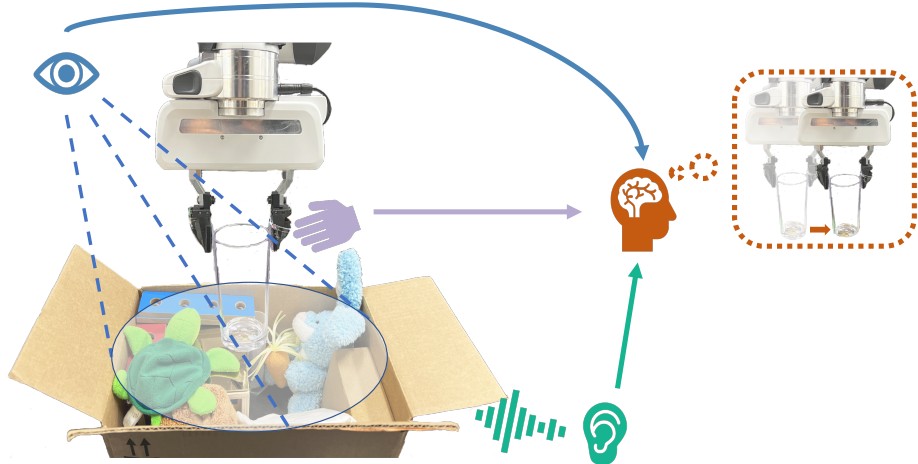

Figure 1: Illustration of the real-world dense packing task that we tackle, where the robot needs to insert a glass into the densely cluttered box by leveraging multisensory feedback.

tative analyses, we uncover interesting findings about the characteristics of the three modalities in robotic tasks that correlate to human behaviors and agree with our intuition.

Our main contributions are threefold. First, we present the first in-depth study that simultaneously leverages vision (from a camera), audio (from a contact microphone), and touch (from a tactile sensor) for robotic manipulation tasks. Second, we introduce a multisensory self-attention model for fusing all three sensory modalities, and strongly outperforms existing approaches. Finally, we show its applications on two challenging robotic tasks: dense packing and pouring, and demonstrate the benefit of multisensory perception in robot learning.

## 2    Related Work

**Deep Multimodal Learning.**    The progress of deep learning algorithms [15, 16, 17, 18, 19] in the last decade has contributed to the the growing potential of learning from heterogeneous sources of data streams, including visual, textural, acoustic, motion, and temporal data. Many real-world problems are inherently multimodal. Research in different fields have combined multiple modalities for various tasks, including representation learning [20, 21, 22, 23], embodied learning [24, 25, 26, 27], video understanding [28, 29, 30, 31, 32], and cross-modality prediction [33, 34, 35, 36, 37]. Different from all the work above, we focus on three important *sensory* modalities—vision, audio, and touch, and investigate their different roles in robotic manipulation tasks.

**Robot Learning for Manipulation.**    Robotic manipulation has been a long-standing challenge [38, 39, 40]. To tackle these challenges, robot learning methods has been widely used in various robotic manipulation tasks [1, 2, 41, 3, 4, 5, 42]. The tasks in this paper are closely related to the classical peg-in-hole insertion task [43, 44] and the pouring task [42, 45]. In contrast to the previous work, we leverage multiple sensory modalities to solve complex tasks such as *dense packing* and *pouring to a target amount*. In order to accomplish these two tasks, the robot needs to focus on different modalities at different stages of the tasks.

**Multisensory Perception for Robotics.**    Recent inspiring work uses audio or touch to enhance a robot's perception in a series of interesing tasks. Audio has been shown to be useful for classifying object instances [46] or terrain types [47], modeling object dynamics [8, 7], estimating amount and flow of granular material [48], learning food embeddings [9], separating impact sounds [49], and mitigating visual occlusion [6]. Tactile sensing is used for shape reconstruction [14, 50, 51], robotic grasping [10, 11, 12], peg insertion [13, 52, 53], dense box-packing [54], planar pushing and door opening [55], and object pose estimation [56].

Despite the encouraging progress, no prior work has simultaneously leveraged visual, acoustic, and tactile sensory data to tackle challenging robotic manipulation tasks. We present the first in-depth

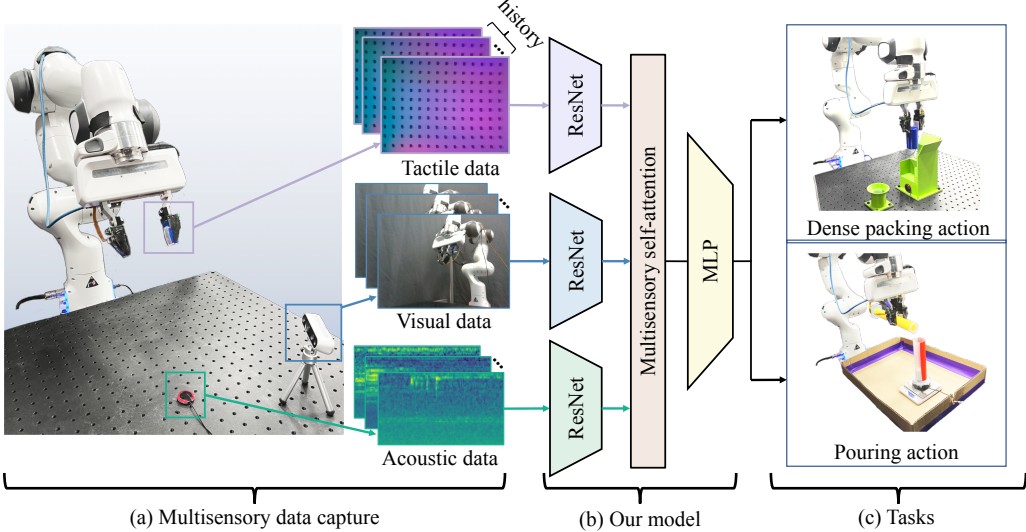

(a) Multisensory data capture  (b) Our model  (c) Tasks

Figure 2: Our multisensory robot learning framework. The visual, acoustic, and tactile data from the corresponding sensors (left) are processed and fused by our multisensory self-attention model (middle) to predict a task-specific action for the dense packing task and the pouring task (right).

study on fusing all three sensory modalities, and systematically explore their respective roles in the context of robot learning. The work most related to ours is ObjectFolder [57, 58], where they also use all three sensory modalities. However, while their goal is to create a dataset of neural objects, and use simulated multisensory data for some simple tasks, we collect data from a real robot system and study two complex real-world robot manipulation tasks.

## 3  Multisensory Robot Learning

We aim to demonstrate how multisensory perception can help solve complex robotic manipulation tasks. In the following, we first introduce how we capture the visual, acoustic, and tactile data and their respective characteristics (Sec. 3.1, Fig. 2a). Then, we introduce our multisensory self-attention model for fusing multiple modalities (Sec. 3.2, Fig. 2b). Finally, we show how we leverage multisensory data for two challenging robotic tasks, i.e., dense packing and pouring (Sec. 3.3, Fig. 2c).

### 3.1  Multisensory Data Capture

At each time step, we record a tuple of observations from the visual ($V$), acoustic ($A$), and tactile ($T$) sensors, respectively. Fig. 2a shows our data capture setup and examples of multisensory observations. Below we introduce the details of how we enable the robot to see, hear, and feel:

- **See:** We capture the visual data using a third-person view camera placed facing the robot, which provides a global view of the robot setup and its end-effector position.

- **Hear:** We attach a piezo contact microphone to the object of interest, and receive acoustic signals through vibration. The microphone is sensitive to any contact with the surface that the microphone is attached to and captures minimal environmental noise.

- **Feel:** We attach GelSight sensors [59, 60] to the fingertips of the robot gripper. GelSight sensors are vision-based tactile sensors that interact with the object through an elastomer and measure the high-resolution contact geometry and forces with an embedded camera.

During data collection, the robot control input **u** is given by the human expert policy, which is selected from a task-dependent discrete action space and commanded in real-time through the keyboard, which we will discuss in Sec. 3.3.

## 3.2 Multisensory Self-Attention Model

Next, we introduce our Multisensory Self-Attention (MULSA) model as shown in Fig. 2b.

**Model Input.** At each timestep, the model takes as an input $\{X_V, X_A, X_T\}$, which correspond to the visual, acoustic, and tactile sensory data captured by the camera, contact microphone, and GelSight sensor, respectively. For audio, we use the mel spectrogram representation to encode the acoustic signal. For each modality, we append a brief history of observations from previous timesteps, making $X_V \in \mathbb{R}^{H_V \times W_V \times 3 \times N}$, $X_A \in \mathbb{R}^{M \times L \times N}$, and $X_T \in \mathbb{R}^{H_T \times W_T \times 3 \times N}$, where $N$ is the number of total observations used, $H_V, W_V$ are the height and width of the visual images, $M, L$ denote the length of the frequency and the time dimensions of the mel spectrogram, and $H_T, W_T$ are the dimensions of the tactile images.

**Sensory Feature Extraction.** We use a ResNet-18 [16] network to extract feature vectors of dimension $D$ from the observations of all three sensory modalities. We modify the input channels of the first convolution layer of ResNet-18 accordingly. Therefore, for each timestep, we have a set of feature embeddings $\{\mathbf{e}_1, \mathbf{e}_2, \ldots, \mathbf{e}_N\}$ from each modality, where $\mathbf{e} \in \{\mathbf{v}, \mathbf{a}, \mathbf{t}\}$ and $\mathbf{v}, \mathbf{a}, \mathbf{t}$ denote the features extracted from the visual, acoustic, tactile data, respectively.

**Modality-Temporal Feature Fusion.** We use a self-attention mechanism to fuse the set of features $\{\mathbf{v}_1, \ldots, \mathbf{v}_N, \mathbf{a}_1, \ldots, \mathbf{a}_N, \mathbf{t}_1, \ldots, \mathbf{t}_N\}$ from the three modalities. We use the standard multi-head self-attention [18], and concatenate the output feature vectors from the self-attention layer to obtain the fused multisensory feature $\mathbf{m} = [\mathbf{o}_1|\mathbf{o}_2|\ldots|\mathbf{o}_{3N}]$, where $|$ denotes the concatenation operation. As shown in Fig. 3, there are three types of attention in this process: **1)** *cross-modality attention*, which allows the features of one modality $\mathbf{e}_i$ to be compensated by the feature $\tilde{\mathbf{e}}_i$ of another different modality at the same time step, **2)** *cross-time attention*, which allows a feature $\mathbf{e}_i$ to attend to feature $\mathbf{e}_j$ at a different time step within the same modality to capture the temporal change, and **3)** *cross-modality and cross-time attention*, which combines the above two types. Finally, the multisensory feature $\mathbf{m}$ is fed to a Multi-Layer Perceptron (MLP), which outputs the prediction $\hat{\mathbf{u}}$.

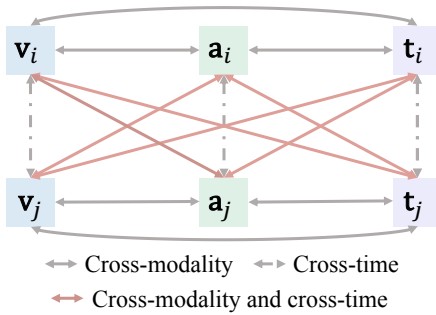

Figure 3: Multisensory self-attention.

**Model Training.** We train the model following the imitation learning pipeline. The model takes as input the time-aligned multisensory data and the human demonstration action $\mathbf{u}$ to learn a policy $\hat{\mathbf{u}} = \pi_\theta(X_V, X_A, X_T)$. All actions are picked from a discrete set defined for both manipulation tasks that we tackle (discussed below). We use the cross-entropy loss between the ground truth action $\mathbf{u}$ and predicted action $\hat{\mathbf{u}}$ for training the network.

## 3.3 Task Setup

Fig. 2c illustrates the settings of dense packing and pouring, which we introduce below.

**Dense Packing.** The goal of this task is to fit an object into a small space in an almost full box. To successfully place the object, the robot needs to first move the object from the starting position to a location above the box, then avoid obstacles and locate the empty space in the box, and finally correctly insert the object. We perform two versions of dense packing: one with 3D printed objects that allow us to do a more controlled study on the role of each modality for this task, and one with diverse real-world objects.

For the first case, we 3D print a cylinder peg as the in-hand object and a base that is covered with two walls as a substitution for a real box. Similar to a real cardboard box, the base has walls on top of it. As shown in Fig. 4a, we design four types of bumps with different configurations inside the base: 1) hard slanted bump, which has a hard 45° slanted surface, 2) soft slanted bump, which has a 45° slanted surface padded with a layer of soft sponge, 3) left flat bump, which is a flat bump located on the left, and 4) back flat bump, which is a flat bump located on the back. For the second case, we set up a more challenging scenario where we use diverse real-world objects for real dense packing as shown in Fig. 1, and the robot needs to fit a glass into a crowded box. In this setup, we

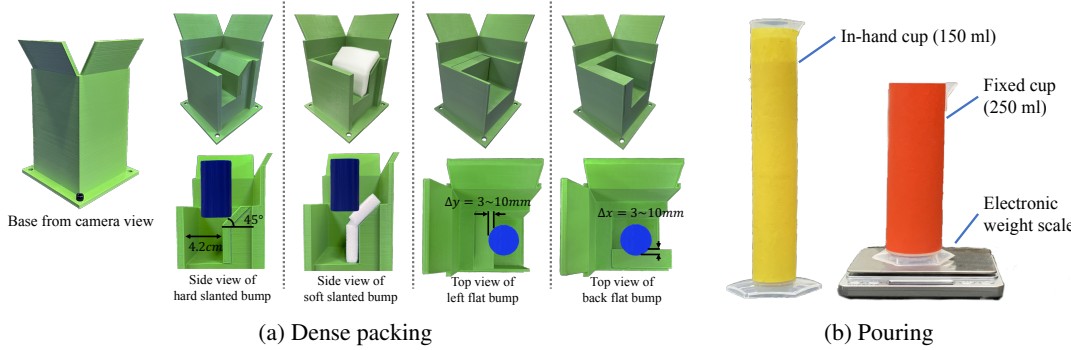

Figure 4: Illustration of our task setup for the dense packing task and the pouring task.

have different objects including a thin metal board, a soft stuffed toy, a towel, and a plastic plate. During the process, the glass will interact with these objects, and the robot must respond differently to avoid the obstacles.

In this task, the robot predicts the $(x, y, z)$ displacement of the end-effector, with each dimension can be positive, negative, or zero. The robot moves along $x, y$ axes to adjust where to insert in the horizontal plane, and vertically along the $z$ axis to insert the object.

**Pouring.** The goal of this task is to pour tiny beads from one cylinder container to the other and reach the target level. To complete the task, the robot first needs to locate and align the two cups, then rotate the cup in hand to start pouring, and stop at the right moment to avoid overflow.

We use small beads of diameters of 1mm to resemble water in the pouring task for ease of experiments. As shown in Fig. 4b, we use two cylinder containers as cups. One cup is grasped by the robot, with either 60g or 100g beads in it initially before pouring. Another cup is fixed on the table to catch the dropped beads. The goal is to pour 40g beads into the fixed cup. The closer the weight of the beads is to the target amount, the better this task is accomplished. An electronic weight scale is placed below the fixed cup for accurate quantitative measurements.

In this task, the robot predicts a 2-dimensional action $(x, \phi)$, where the horizontal movement $x$ is required to align the two cups, and $\phi$ is the rotation angle that corresponds to the pouring action. Same as the dense packing task, the action in each dimension can be positive, negative, or zero.

See Supp. for details of the setup and how we collect human demonstrations for these two tasks.

## 4 Experiments

We validate the benefit of multisensory perception for robotic manipulation on two challenging tasks: dense packing and pouring.

**Baselines:** We compare our model with the following baselines:

- **Direct Concat:** A baseline method that directly concatenates the extracted feature vectors of all time steps from all three modalities. This is commonly used in the literaure [13, 52, 10, 57, 58] for modality fusion, making it a broadly representative baseline for standard practice.
- **Du et al. [6]:** A recent audio-visual imitation learning method that uses audio to mitigate visual occlusion. We adapt their LSTM-based model to fuse the three modalities for action prediction.
- **MULSA (V):** A variant of our model that only uses visual sensory data (V).
- **MULSA (V+A/V+T):** Two variants of our model that only use two of the sensory modalities. The model complements vision with either audio (V+A) or touch (V+T).
- **MULSA (A+T):** A variant of our model that only learns over audio and touch, and adopts a heuristic visual policy to complete the aligning process during the task.

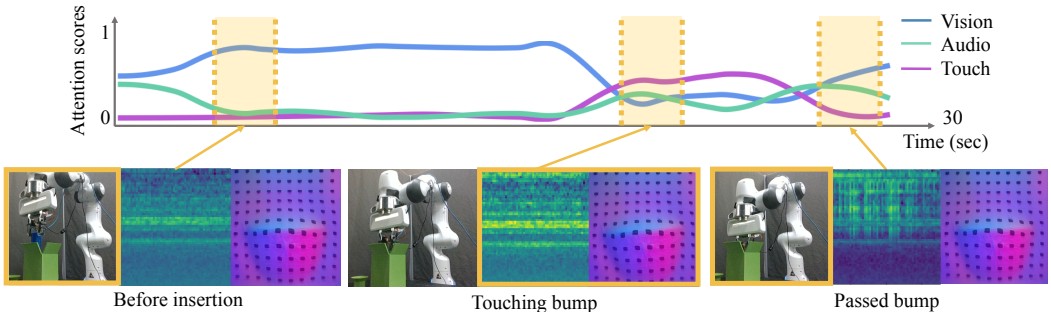

(a) Sensory observations of three representative moments for dense packing: no contact happens before insertion (left), the peg touches the bump and generates informative acoustic and tactile signals (middle), and the peg has passed the bump and just needs to be pushed down (right).

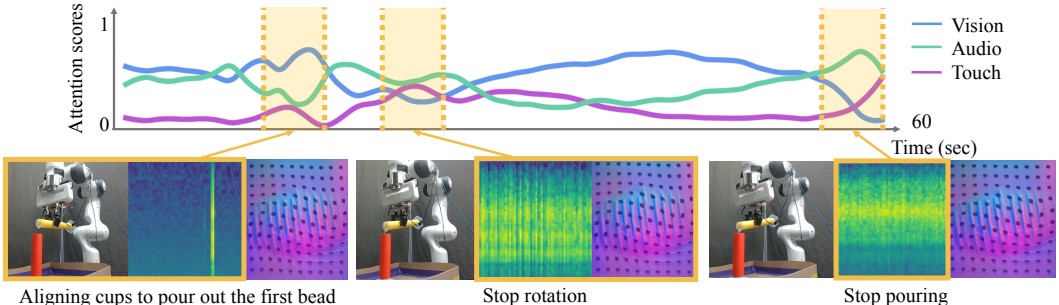

(b) Sensory observations of three representative moments for pouring: the robot aligns the two cups and the very first bead drops into the cup (left), the robot stops rotating the held cup to pervent beads from overflowing (middle), and the cup is retrieved to end pouring (right).

Figure 5: Visualization of the aggregated attention scores for each modality as the robot completes (a) the dense packing task and (b) the pouring task in two test trials.

## 4.1 Physical Setup and Implementation Details

**Robot Setup.** We run all experiments on a Franka Emika Panda Arm, which is equipped with an inverse kinematics solver that maps 6-DoF Cartesian space displacement command input to the 7-DoF joint action. For both tasks, we generate Cartesian space displacement commands at a policy frequency of 10 Hz. Then the low-level control loop maps the displacement to the force and torque input in the joint space to the robot at 200 Hz.

**Data Collection.** As shown in Fig. 2a, the visual data are recorded by an Intel RealSense D435 camera with a resolution of $320 \times 240$ at a frequency of 60Hz. The acoustic data are recorded with a HOYUJI TD-11 piezo-electric contact microphone at a sampling rate of 44.1kHz. The microphone is connected to a TASCAM US-4×4HR audio interface, which transmits the signal to a workstation. The tactile data are recorded by a GelSight [59] sensor with a resolution of $400 \times 300$ and a frequency of 30Hz.

**Model Details.** We implement our multisensory self-attention model in PyTorch. We downsample the visual and tactile frames to dimension $140 \times 105$, and randomly crop regions of size $128 \times 96$ during training. For audio, we resample the received signal from our piezo-electric contact microphone at 16kHz. The short-time Fourier transform is computed using a Hann window of length 25ms, hop length of 10ms, and mel size of 64 to generate a mel spectrogram of size $64 \times 50$ for a 0.5s audio segment at each time step. We use $N = 6$ observations for all three modalities. The 6 frames of the visual and tactile obseravtions span a time interval of 3 seconds to align with the audio input. See Supp. for details of our model architecture.

## 4.2 Dense Packing

We collect 10 episodes of human demonstrations for dense packing for each of the four designed bases (hard slanted, soft slanted, back flat, left flat). We train a single policy for all four scenarios

|                     | Hard slanted | Soft slanted | Back flat | Left flat | Avg. succ. rate |
|---------------------|:------------:|:------------:|:---------:|:---------:|:---------------:|
| Direct Concat       | 0.90         | 0.50         | 0.00      | **1.00**  | 0.60            |
| Du et al. [6]       | 0.50         | 0.30         | 0.50      | 0.60      | 0.48            |
| MULSA (V)           | 0.00         | 0.00         | 0.00      | **1.00**  | 0.25            |
| MULSA (V+A)         | 0.80         | **1.00**     | 0.10      | 0.50      | 0.60            |
| MULSA (V+T)         | 0.00         | **1.00**     | **1.00**  | **1.00**  | 0.75            |
| MULSA (A+T)         | 0.20         | **1.00**     | **1.00**  | **1.00**  | 0.80            |
| MULSA (ours, V+A+T) | **1.00**     | **1.00**     | **1.00**  | **1.00**  | **1.00**        |

Table 1: Dense packing results using 3D printed objects. We report the success rate for testing on each type of the designed four bases and the average performance.

and run 10 test trials on each base with random starting locations. A trial is considered successful if the robot can pass the bump and insert the peg into the base.

Table 1 shows the results. our model MULSA using all three modalities outperforms all the baselines and achieves a 100% success rate in all four scenarios. In contrast, MULSA with vision (V), vision and audio (V + A), and vision and touch (V + T) only achieve an average success rate of 25%, 60%, 75%, respectively, demonstrating the usefulness of each modality and the benefit of multisensory perception. Particularly, from this ablation study, we see that the visual and tactile modalities turn out to be sufficient for the model to learn the correct policy on the flat bumps. The tactile sensor can easily distinguish these different in-hand object poses, therefore extra information in the audio is unnecessary. However, we observe that the model cannot tell the difference between the hard and the soft slanted bumps if not using audio, because the two bumps produce very similar visual and tactile feedbacks, while the sound generated when contacting the hard surface can easily distinguish them. On the other hand, the tactile signal can be very important to distinguish the two flat bumps. When pushed onto these two bumps, the peg tilts in different directions, revealing critical local geometry to avoid the obstacles. When the model only learns over audio and touch and adopts a heuristic visual policy that moves the end-effector to a spot very close to the base, the major failure mode observed occurs on the hard slanted bump, where the robot successfully avoids the bump but fails to continue inserting downward after the signal disappears in audio and touch. Our model also has large gains compared to Direct Concat and Du et al. [6], showing the power of our multisensory self-attention model for modality fusion.

Our model uses a self-attention mechanism to dynamically attend to useful modalities at each time step, and fuses information from the multisensory data. In Fig. 5a, we illustrate the change of the aggregated attention scores of the three modalities in one test trial as the robot inserts the peg into the base with a hard slanted bump. At the beginning of the trial, the peg location is randomly initialized, therefore the model focuses on vision to locate the correct position to insert. After the peg is partially inserted into the base, vision suffers from occlusions by the walls of the base, shifting the model's attention towards both audio and touch. Specifically, there is an obvious contact sound, alerting that the peg is touching a hard bump. Then touch tells how the peg is tilting, helping the model to infer in which direction the robot should move next to avoid the bump. At the end of the trial, the peg has already passed the bump, so the attention is back to vision to move the robot arm downward. See Supp. video for more qualitative examples.

The controlled experiments with 3D printed objects above help us analyze the characteristics of each modality in these different settings. Table 2 shows our results on dense packing using real-world objects as shown in Fig. 1 to further test the effectiveness of our model and demonstrate the power of multisensory perception. Note that this is a challenging setting due to the diverse set of objects of different shapes and material properties. One major failure mode observed in this setting is that the robot cannot recognize whether the obstacle is hard or soft, thus pushing too hard onto a hard surface and getting stuck. When touching hard surfaces, such as the plastic plate, the robot should move to avoid breaking them. If the object is soft, such as the toy or the towel, the robot should squeeze down because they can deform. Another typical failure mode of the baseline models is that the robot misses the correct range of the empty space in the box during the initial aligning phase, and the glass is moved out of the box. Still, our model consistently outperforms all the baselines.

| Methods | Succ. rate |
|---|---|
| Direct Concat | 0.60 |
| Du et al. [6] | 0.30 |
| MULSA (ours) | **0.80** |

Table 2: Dense packing results with real-world objects.

| Initial weight (g) | 60 | 100 | Average |
|---|---|---|---|
| Direct Concat | $2.24 \pm 1.34$ | $2.69 \pm 1.85$ | $2.46 \pm 1.54$ |
| Du et al. [6] | $7.35 \pm 3.59$ | $2.70 \pm 2.19$ | $5.02 \pm 3.72$ |
| MULSA (V) | $20.87 \pm 0.0$ | $60.87 \pm 0.0$ | $40.87 \pm 0.0$ |
| MULSA (V+A) | $8.89 \pm 0.96$ | $21.51 \pm 6.33$ | $15.20 \pm 7.90$ |
| MULSA (V+T) | $2.81 \pm 1.30$ | $8.35 \pm 1.14$ | $5.58 \pm 3.14$ |
| MULSA (V+A+T) | $\mathbf{1.06 \pm 0.83}$ | $\mathbf{1.76 \pm 1.16}$ | $\mathbf{1.41 \pm 1.02}$ |

Table 3: Mean weight error of beads poured into the cup (mean $\pm$ standard deviation) (g).

### 4.3 Pouring

We collect 10 human demonstrations with two initial settings: 60g or 100g beads held in the cup. The goal is to pour 40g beads into the fixed cap. We train a single policy for these two settings and run 10 test trails for each setting with the fixed cup slightly shifted from the training position.

Table 3 summarizes the results. We show the mean weight error of the beads poured into the fixed cup compared to the target weight. MULSA with only vision (V) always keeps pouring all beads into the fixed cup since it has no clue about the current status due to the highly similar visual feedback across the entire trial. With the help of one extra modality (V + A or V + T), the model obtains large gains. Our MULSA model using all three modalities achieve the best results, outperforming the two baseline methods that use a different mechanism for fusing the three modalities.

In Fig. 5b, we visualize three key moments when our model attends to different modalities. When pouring is about to begin, the model attends to vision to align the two cups, with some attention on audio to capture the sound of the first few dropping beads. Later, more beads start to drop and make sounds in the fixed cup. The pitch of the sound increases as more beads are poured into the cup. The center of mass of the in-hand cup also begins to vary, making changes to the gel deformation of the GelSight sensor. Therefore, the model's attention scores are gradually shifted to audio and touch. To prevent the overflow of the beads, the robot needs to stop rotation and transit to a static pose at the right moment. The last critical moment is to predict when to stop pouring, which heavily relies on the acoustic feedback as shown in the last example. This also agrees with our experience when testing without the audio feedback—We are more likely to see a lag in prediction at the end of the whole pouring process, leading to more beads being poured out than desired.

## 5 Conclusion and Limitations

We presented a multisensory robot learning framework that leverages visual, acoustic, and tactile sensory data to solve two challenging manipulation tasks: dense packing and pouring. Our multi-sensory self-attention model strongly outperforms a series of baseline methods by effectively fusing and filtering out the most useful information from the three sensory modalities. We have the following observations for the roles of vision, audio, and touch for robotic manipulation tasks: Vision usually provides general positional information, yet it often suffers from occlusion. Audio provides immediate feedback and also indicates the characteristics of the objects such as material properties, but is less informative when there is no contact. Touch captures in-hand object dynamics, which best indicates the local geometry as well as the direction and magnitude of the contact force. We hope our work can inspire new research to use multisensory perception for robot learning.

**Limitations:** Our current framework uses behavioral cloning for learning the expert policy, which requires collecting human demonstrations. Though we have shown encouraging results using imitation learning, the way a robot perceives multisensory signals might be different from how we humans leverage them during demonstrations. It would be interesting future work to use reinforcement learning to have the robot automatically uncover the best way of fusing multisensory data streams and tackle more challenging manipulation tasks. Another potential improvement is to establish a more general robot setup that can be easily adapted to more general tasks. For example, we can design a tailored robot gripper that has the contact microphone and the GelSight sensor built into it to receive the acoustic and tactile signals simultaneously in any robot manipulation tasks.

**Acknowledgments**

We thank Chen Wang, Kyle Hsu, Yunzhi Zhang, and Koven Yu for helpful discussions or feedback on paper drafts. The work is in part supported by the Stanford Institute for Human-Centered AI (HAI), the Toyota Research Institute (TRI), NSF RI #2211258, ONR MURI N00014-22-1-2740, Adobe, Amazon, Meta, and Samsung.

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
