# OpenReview forum: "See, Hear, and Feel: Smart Sensory Fusion for Robotic Manipulation"
_robot-learning.org/CoRL/2022/Conference — CoRL 2022 Poster_

### Official Review · Reviewer_2ByC · 2022-07-18

**Originality:** Very Good
**Technical Quality:** Excellent
**Clarity Of Presentation:** Very Good
**Impact:** 4

**Recommendation:**

Weak Accept: I recommend accepting the paper, but will not argue for my recommendation if the majority of other reviewers have a different opinion.

**Summary:**

This work proposes a new multisensory self-attention model for fusing visual, acoustic, and tactile sensory data, and uses imitation learning for action prediction. The model uses a self-attention mechanism to attend both across modalities and across time, outperforming prior fusion methods. They use a self-attention mechanism to fuse the set of features using a standard multi-head self-attention architecture.



**Issues:**

- I did not come across any discussion with regards to sensor calibration and synchronizing across multi modality - Did the authors find this to be an issue? Or was combining all the sensors a smooth process in general?
- There is reference to a method called `mel spectrogram representation` - I'm not familiar with this, perhaps the authors can provide a reference and explanation of how this data is generated.

**Quality Of The Limitations Section:**

Limitations are addressed clearly

**Reviewer Expertise:**

4: The reviewer is confident but not absolutely certain that the evaluation is correct

**Robotics Focus:**

Sufficient demonstration on hardware

**Strengths And Weaknesses:**

### Strengths
- The use of multi-sensory inputs to perform robotics tasks
- Some level of engineering required to build and integrate the sensors and synchronize the data acorss multi modality
- The results are compeling and the evaluations of the effect of adding each modality is also convincing
- There is a lot of detail about their hardware setup, their 3D printed task items etc.
- The use of multi sensory inputs in this domain is rather new and more work is exploring this area.

### Weaknesses
- The architecture on its own is very simple, its just a simple feature extraction using ResNet and feature fusion using multi-head self attention architecture.
- The paper mostly contains engineering detail of the task setup (This is still useful but this is more of an engineering paper).


**Summary Of Recommendation:**

I definitely like the idea of applying sensory fusion techniques to utilize multiple sensors. The experimental setup and the results are convincing. My only thought is that, the method on its own, is not novel, its just a simple architecture for combining multiple modality. Even though it does establish a baseline to further explore other ideas to improve upon.
Finally, this paper for me is more of a well-executed robotics engineering paper, as opposed to strong core ideas being proposed here.
I am also not familiar with related work in this domain, but again combining multiple sensors for this particular manipulation domain is new but I imagine there might be more related work combining image and touch sensors.

---

> ### Author Response · Authors · 2022-08-24
> **Response to Reviewer 2ByC**
>
> Thank you for the constructive feedback!
>
> #### **1) The simplicity of the architecture.**
>
> The model we propose (MULSA) is simple in its architecture, yet we find it an effective solution to the multimodal sensor fusion manipulation problem. As shown in Fig. 5 in the main text, the attention mechanism allows the model to focus more on the useful information embedded in different modalities, and the attention scores keep varying depending on the current status of the task.
>
> We actually have experimented with another network architecture on the dense packing and pouring tasks. The experimented model uses a ResNet encoder with a Transformer network, which segments the encoded sensor inputs into small patches and then performs multi-headed self-attention over them. However, this model turns out to suffer from severe overfitting and has a much lower success rate than our MULSA model on all tasks. We suspect that this model is more data hungry and needs a significantly larger dataset to train, while our MULSA model is simple yet effective and more sample efficient.
>
> #### **2) A well-executed robotics engineering paper?**
>
> Thank you for raising this very good point. We agree that a major contribution of our submission is a well-executed robotics system, which we believe is valuable to the robotics community by nature, especially considering CoRL itself is a robotics venue.  We also believe that our submission is making a conceptual contribution. As we summarize in Sec. 1 of the main text, the core conceptual idea in our work is that we should always attempt to incorporate the three sensory inputs for robot manipulation, and that their effective fusion provides more meaningful information than that implicitly embedded in each. It is to prove this statement that we have designed and engineered a novel robotic setup, performing an in-depth analysis of the role played by each of the modalities in different tasks.
>
> We would like to further highlight that our significant engineering effort is valuable:  although some prior works demonstrated promising results of using one or two modalities on different tasks (Liang et al., 2019, Lee et al., 2019, Dong et al., 2019), we are the first to combine and learn over all three modalities (vision, audio, and touch); therefore, the hardware requires careful design and tuning. We have shown in Sec. 4.2 and Sec. 4.3 that with our proposed method, the three sensory inputs are effectively fused, and different modalities play more critical roles as the task proceeds. Based on these observations, we show that vision is important for the general positional relationship, while audio and touch provide more fine-grained local information about material properties and geometry, respectively. Please see Sec. 5 of the main text for the complete conclusions we make about the main idea of our work.
>
> #### **3) Related work combining image and touch sensors:**
>
> Yes, there are several works that combine vision and touch, as discussed in our related work section. The work that is most related to ours is ``Making sense of vision and touch”, which studies the peg insertion task using a table-mounted camera with the force-torque sensing of the robot arm. To demonstrate the effectiveness of our approach, we perform another set of new experiments on the dense packing task where we train the MULSA model with the inputs from vision and the readings from the force-torque feedback sensor, and we compare the performance with our original method of combining vision, audio, and touch. In this experiment, we substitute the audio and tactile sensors with the Franka arm's in-built force-torque (FT) sensor. At each time step, the FT sensor returns 7 torque values corresponding to the 7 joints of the robot. We stack the FT readings recorded within a short time history and encode this observation with 4 layers of 1D convolutional network.
>
> The average test performance of MULSA using the table-mounted camera with the force-torque sensor is only 0.25 on the four bases. The results show that the force-torque sensor does not provide sufficient information for the robot to complete the task. This is because audio and touch can provide more fine-grained information about the local geometry, while the force-torque sensor can only reflect a meaningful change when the contact force between objects gets stronger, for example, when the peg is pushed very hard onto the surface. Therefore in tasks that require more sensitive and precise feedback, which is common in our daily life as our packing and pouring tasks, our solution can better capture different characteristics about the object contacts and reach a higher performance. Please refer to the revised supplementary material for more detailed information about the setup, results, and analysis about the experiment.

---

> > ### Author Response · Authors · 2022-08-24
> > **Response to Reviewer 2ByC (continued)**
> >
> > #### **4) Discuss with regards to sensor calibration and synchronizing across multiple modalities:**
> >
> > As pointed out in the review, we put much effort into synchronizing across multiple modalities. To ensure all modalities correctly correspond to the current action step, our solution involves modifications in both the sensor input streaming scripts and our main control loop. As discussed in Paragraph 2 of Sec. 4.1 of the main text, we first examine and ensure that the camera and GelSight streaming frequencies (60 Hz and 30 Hz, respectively) are sufficient for data collection. In the control loop, we set a 10 Hz clock to retrieve the most recently recorded frame from visual and tactile sensors at the same time and timestamp them. The audio sensor keeps recording during the entire experiment at 44100 Hz, and we can easily retrieve the audio segment that align with the other two modalities by checking the timestamp.
> >
> > #### **5) Reference and explanation of how the mel spectrogram representation is generated.**
> >
> > The raw acoustic signal we record is a sequence of data across time at the sampling frequency of 44100 Hz. When processed through the mel spectrogram, the signal is first segmented into small windows, and then each window is decomposed into frequency and magnitude components through the fast Fourier transform. In this way, we obtain a spectrogram that disentangles time and frequency in the acoustic signal. Finally, the frequency axis of the spectrogram is converted from the linear scale into the mel scale, which helps humans process the high and low frequencies more easily. Mel spectrogram is commonly used in prior work  (Henshey et al. 2017, Sawhney et al. 2021, Gao et al. 2022) for processing audio information.
> >
> > Furthermore, to further help to understand the mel spectrogram representation, we have newly performed an additional ablation experiment on the pouring task to analyze the roles of magnitude and frequency in the mel spectrogram representation when using audio for the pouring task. We only keep the information in one component between magnitude and frequency, and eliminate the other through normalization and averaging. There are initially 60g of beads initially in the cup. Results are shown in the table below.
> >
> > |              | V+A (mag only)+T |  V+A(freq only)+T | Ours, V+A(mag+freq)+T  |
> > |    :---:    | :---:  |   :---:   |   :---:  |
> > | Mean error (unit: g) | 7.50 | 19.58  |  1.06  |
> >
> > From the experiments, we observe that the frequency information is more important than the magnitude on this task, yet the latter still compensates for the missing piece and helps the robot perceive the environment more easily. This also aligns with our daily observation that when we pour water into a cup, we hear the pitch getting higher as the water level rises, with a more subtle change in the magnitude of the sound. The results and analysis of this experiment are summarized in Sec. 8 of the supplementary material.

---

### Official Review · Reviewer_329W · 2022-07-30

**Originality:** Very Good
**Technical Quality:** Very Good
**Clarity Of Presentation:** Very Good
**Impact:** 4

**Recommendation:**

Strong Accept: I recommend accepting the paper and will argue for my recommendation even if other reviewers hold a different opinion.

**Summary:**

The paper proposes a robot system that fusing three sensory modalities, namely seeing, hearing, and touching, with a self-attention model. The system is deployed to accomplish two challenging tasks, dense packing and pouring. The hardware experimental results show that the robot system with such fusion of three modalities outperforms the systems with only two modalities or without the self-attention mechanism.

**Issues:**

1.	The authors are suggested to add the variations in addition to the average performance in Table 1, 2, and 3 to show the consistency of the proposed robot system.

**Quality Of The Limitations Section:**

Limitations are addressed clearly

**Reviewer Expertise:**

4: The reviewer is confident but not absolutely certain that the evaluation is correct

**Robotics Focus:**

Sufficient demonstration on hardware

**Strengths And Weaknesses:**

The main strengths of the paper are:
1.	The training and testing of the self-attention model are done with real hardware, which makes the conclusions more convincing.
2.	The experimental setup are well explained with relevant details.
3.	The supplementary video is helpful in demonstrating the qualitative performances of the robot system.

The main weaknesses of the paper are:
1.	Some results of the validation experiments are not very intuitive to understand. The authors need to discuss a bit more to make the results more meaningful to the readers. For example, in the dense packing tasks, the MULSA (V+A+T) method outperforms the MULSA (V+T) only in the Hard slanted case. Seems it’s always the case if the robot is able to obtain more information about the environment, it is more likely to distinct different situations, given that it uses more or less the same algorithms.
2.	It would be more convincing if the authors could open source their codes in the near future for reproduction of their results in the community.

The paper shows promising results in ultilizing smart sensor fusion with a self-attention model, which is inspiring for other researchers in the community.

**Summary Of Recommendation:**

The paper presents a concise yet effective design of multi-modal sensory fusion and impress the readers with interesting demonstrations in real robotic manipulation tasks. The idea is interesting to the community and the presentation is well organized. Hence, I recommend accepting the paper.

---

> ### Author Response · Authors · 2022-08-24
> **Response to Reviewer 329W**
>
> Thank you for the constructive feedback!
>
> #### **1) Intuition of the validation experiments. MULSA (V+A+T) outperforms MULSA (V+T) only in the hard slanted case for dense packing.**
>
> The relevant results appear in Table 1 of the main text, where we show MULSA (V+T) is reaching 100% success rate on three of the four settings, while MULSA (V+A+T) only outperforms this baseline in the hard slanted case. The reason behind this is two-fold. First, the visual and tactile modalities turn out to be sufficient for the model to learn the correct policy on the flat bumps. As shown in Fig. 4a of the main text, the two flat bumps differ in their locations, which results in very different peg poses upon contact. The tactile sensor can easily distinguish these different in-hand object poses, therefore extra information in the audio is not necessary for the robot to take the correct action. On the other hand, as we already discussed in Sec. 4.2, we see that the model cannot learn the proper strategy on the two slanted bumps without audio, and the experiment result shows that it is overfitting to the soft slanted case and tries to adopt the same strategy when the surface is actually hard. This is because the tactile sensor can capture little meaningful information about the surface material, so it is difficult for it to differentiate between the two surfaces that share highly similar geometry. When this missing piece about the environment is compensated through the third modality, audio, the model is able to learn the correspondence between the environment and the action in a more comprehensive way.
>
> We hope the explanation clarifies your confusion about the intuition behind our experiments. We appreciate your suggestion, and we have added some of these discussions to Sec. 4.3 of the main text.
>
> #### **2) Seems it's always the case if the robot is able to obtain more information about the environment, it is more likely to distinct different situations?**
>
> Although we see the test performance improving as we use more modalities in our ablation study among vision, audio, and touch, this is not always the case.  Some prior works have proposed solutions to other similar manipulation tasks with visual and force-torque feedback, such as the method used in ``Making sense of vision and touch" (Lee et al., 2019), which tries to tackle the peg insertion task. To demonstrate the effectiveness of our approach, we perform another set of new experiments on the dense packing task where we train the MULSA model with the inputs from vision and the readings from the force-torque feedback sensor, and we compare the performance with our original method of combining vision, audio, and touch. In this experiment, we substitute the audio and tactile sensors with the Franka arm's in-built force-torque (FT) sensor. At each time step, the FT sensor returns 7 torque values corresponding to the 7 joints of the robot. We stack the FT readings recorded within a short time history and encode this observation with 4 layers of 1D convolutional network.
>
> The average test performance of MULSA using the table-mounted camera with the force-torque sensor is only 0.25 on the four bases. The results show that audio and touch provide more helpful information than the force-torque sensor to the robot. The main reason behind this is that the audio and tactile sensors are highly sensitive to the local dynamics, and any change can be reflected in immediate changes in the signals. Although the robot has access to extra information in this case, the performance on the task is not as good as the model that fuses vision, audio, and touch, as the latter provides more fine-grained information about the environment that cannot be easily perceived by many other sensory inputs. The results suggest that just more information does not necessarily help, but it is intelligently fusing the right modalities that help the robot to succeed in the task. Please refer to the revised supplementary material for more detailed information about the setup, results, and analysis about the experiment.
>
> #### **3) Open source the codes:**
> Yes! We will release all data and codes upon publication.
>
> #### **4) Add the variations in addition to the average performance in Table 1, 2, and 3 to show the consistency of the proposed robot system:**
>
> Thanks for the advice. For Table 1 and Table 2 of the main text, where we show results on the dense packing task, the performance is computed as the number of successful trials out of 10 test trials, therefore we did not define the standard deviation. Table 3 of the main text and Table 1 of the supplementary material show results on the pouring task, and we have now added the standard deviation values for each mean error. From the results, we see that our model consistently achieves better performance across multiple trials on different settings.

---

### Official Review · Reviewer_yYyH · 2022-08-01

**Originality:** Good
**Technical Quality:** Very Good
**Clarity Of Presentation:** Excellent
**Impact:** 3

**Recommendation:**

Weak Accept: I recommend accepting the paper, but will not argue for my recommendation if the majority of other reviewers have a different opinion.

**Summary:**

The main contribution of the paper is the system setup of using 2 gelsight finger tips, 1 contact microphone, and a RealSense D435 camera looking at the robot to enable the robot to accomplish 2 tasks: placing an item into a cluttered box / putting a peg in different 3d printed boxes to simulate different packing scenarios, and pouring beads from one graduated cylinder to another. They way they accomplish this is by feeding all 3 data streams into individual resnet18 networks which are then concatenated and fed through a self-attention model that finally goes into a MLP that is used to control the robot. The main way they train the robot's policy is through collecting human demonstrations by allowing them to move the robot in delta movements and tilting the cylinder to start pouring.

**Issues:**

So as I said in my strengths and weaknesses, I feel like there are some issues that need to be addressed by a change in the setup where you could use a wrist-mounted Realsense D435 camera and the Franka Robot's in-built force sensing to try and also complete the first peg in the box task. In addition, I feel like an external non-contact microphone would be useful in sensing pouring even without vision.
However, I don't think it would be fair to you as an author to have to redo all the experiments by switching the setups. I just wanted to warn you that the paper would be much stronger if the robot sensor setup didn't have to be customized for every different situation. Otherwise we would be approaching the point of automation where if we wanted to be extra precise, you could have put an overhead camera to estimate the mesh of the box and do the planning for the cluttered box placing just like in industrial pick and pack startups or using a overhead laser and usb-connected scale in order to precisely pour the exact amount desired.

So basically, I want you to discuss these limitations in your actual limitations section instead of just talking about needing human demonstrations instead of using pure reinforcement learning in your current limitations section.
I think you did contribute something with your setup and self-attention model; however, I think the tasks that were chosen to demonstrate the efficacy of the system were not really ideal to show the reason why you need all the different modalities. It would have been better if you had done some like previous cargo transport like graham crackers. That would show off why using the tactile sensors would prevent the robot from closing the gripper all the way in order to not break the graham cracker and a microphone could detect when the graham cracker crumbles. The vision would be helpful for locating the graham cracker for the initial grasp location.

**Quality Of The Limitations Section:**

Additional details required

**Reviewer Expertise:**

5: The reviewer is absolutely certain that the evaluation is correct and very familiar with the relevant literature

**Robotics Focus:**

Sufficient demonstration on hardware

**Strengths And Weaknesses:**

I think the combination of all the sensors into a combined system is a good engineering effort. Using the self-attention mechanism also makes sense to allow the robot to figure which modality would be most helpful at a certain point in the task. However, I am not really sure if the experiments were that noteworthy. If the authors had used the Franka robot's force sensing and some simple impedance control, would they have been able to achieve similar performance in the peg in box task? In addition, I have seen a couple of other interesting papers of pouring liquids into containers by using a non-contact microphone because it would be a pain to have to attach a contact microphone to each object. The paper I am referencing is "Making sense of audio vibration for liquid height estimation in robotic pouring" because it is true that when water fills up in certain bottles while I am refilling it, the sound will change as it rises and I can use that to figure out when to stop. Why was the contact microphone not attached to the robot instead so it could transfer across both tasks without changing the setup between the 2 tasks? Is it because the contact microphone would not be able to pick up sounds due to dampening between objects due to the gelsight sensor? Basically, I think that the experiments shown in this paper do show the useful properties of incorporating all 3 sensor modalities. However, I would have instead liked for the robot to be fully sensorized instead, so that changing tasks would not have required any significant differences. Also I wasn't sure where the contact microphone was located under the actual cardboard box during that task or if it was attached to the side because in the video, you first showed that the contact microphone was placed onto the table surface, but when I looked closely at each task, the contact microphone was attached to the 3d printed boxes, which transmits the sound well, but different materials might transmit the sound differently and may not work as well. Also the contact microphones were attached under the target graduated cylinder for the pouring task, which is again different. Also I feel like the limitations section was a bit short with just a basic description about using human demonstrations that are not optimal. I feel like you could have written some more concrete limitations about how in certain tasks, these extra modalities may not be useful as well if there is widespread occlusion from the static camera. I feel like it would have been better to also have a robot mounted wrist camera to provide additional information for the peg in the box task. It is way more useful than putting the camera outside the box where of course there will be occlusion, so I think that by not putting the camera on the wrist, the problem got inherently harder in order to demonstrate the efficacy of using the other sensors, but I think that if you had used the robot's force sensors, and a wrist camera, the task would have been much more easily solved.
Another minor critique is that the video was a bit too long. It was 13 minutes. Next time, please keep it short like around 5-8 minutes.

**Summary Of Recommendation:**

Overall my recommendation is weak accept. I think that even though the authors used ResNet18 coupled with a self-attention model and traditional sensors (Gelsight, Realsense D435, and contact microphones), they were able to display how their method did improve on just a direct concatenation method as well as in the ablation studies. However, I still think there are many improvements that can be made in future work regarding sensor placement for actual reuse between a wide range of tasks. If other reviewers think that there was not enough originality in the work for CoRL, then I would agree with them for a weak reject. Overall, I would not mind if this paper were to be accepted at CoRL, but I do not feel strongly either way.

---

> ### Author Response · Authors · 2022-08-24
> **Response to Reviewer yYyH**
>
> Thank you for the constructive feedback!
>
> #### **1) Would using the Franka robot's force sensing and some simple impedance control achieve similar performance in the dense packing task?**
>
> In our experiment, we focus on perceiving the environment through sensory modalities as humans—vision, audio, and touch, and we believe these modalities provide more fine-grained information for robot manipulation tasks. As we mentioned in Sec. 2 of the main text, some prior works have proposed solutions to other similar manipulation tasks with visual and force-torque feedback, such as the method used in ``Making sense of vision and touch" (Lee et al., 2019), which tries to tackle the peg insertion task. To validate the performance of using the vision and force sensing on our task, we perform another set of experiments on the dense packing task where we train the MULSA model with the inputs from vision and the readings from the force-torque feedback sensor, and we compare the performance with our original method of combining vision, audio, and touch. In this experiment, we substitute the audio and tactile sensors with the Franka arm's in-built force-torque (FT) sensor. At each time step, the FT sensor returns 7 torque values corresponding to the 7 joints of the robot. We stack the FT readings recorded within a short time history and encode this observation with 4 layers of 1D convolutional network.
>
> The average test performance of MULSA using the table-mounted camera with the force-torque sensor is 0.25 on the four bases. The results show that audio and touch provide more helpful information than the force-torque sensor to the robot. The main reason behind this is that the audio and tactile sensors are highly sensitive to the local dynamics, and any change can be reflected in immediate changes in the signals. However, the force-torque sensor only shows a change when the contact force gets stronger. As shown in Fig. 1 of the additional supporting material, the force and torque readings do not demonstrate any noticeable changes when the contact happens, yet this moment can be captured better with audio and touch. Please refer to the additional supporting material for more detailed information about the setup, results, and analysis about the experiment.
>
> #### **2) Why was the contact microphone not attached to the robot? Is it because of the dampening effect between objects due to the GelSight sensor? Where was the contact microphone located?**
>
> The contact microphone is not attached to the robot due to two reasons. First, as pointed out in the review, the dampening effect between the grasped object and the robot fingers are exaggerated due to the GelSight sensors, therefore the microphone cannot reflect the vibrations caused by the actual interactions of interest. Second, the contact microphones also capture too much noise when directly attached to the robot since the arm generates noises during normal operation.
>
> In the dense packing task, the microphone is attached to the outer wall of the box as shown in Fig. 2c of the main text. This is also true in the real-world packing scenario, where the microphone is attached to the large cardbox. In the pouring task, the microphone is attached to the bottom of the fixed cup because most surfaces of the cup are curved and do not allow for a secure attachment. In all situations, we do not put extra effort into picking a specific location and it is proved that our model works regardless of where the microphone is placed as long as it maintains a stable and firm contact with the objects. However, we agree that a more general setting that does not require replacing any external devices is better than a customized one, and this would also be an interesting design goal in our future work.

---

> > ### Author Response · Authors · 2022-08-24
> > **Response to Reviewer yYyH (continued)**
> >
> > #### **3) An external non-contact microphone as in "Make sense of audio vibration for liquid height estimation in robotic pouring" would be useful in sensing pouring even without vision:**
> >
> > Thanks, we have added this paper in related work. This paper also studies the pouring process by listening to the sound, and the authors use an external microphone placed close to the cup, while in our work we attach a contact microphone to the bottom of the cup. The main reason behind this is that the contact microphone is better at picking up vibrations of certain objects. In the dense packing task, we have the peg touching different parts of the box, and during the pouring task, the sound of the beans hitting the cup is important to learning. While it is possible to build an ideal anechoic chamber to carry out these experiments, we consider using the contact microphone as a simple and proper tool to eliminate irrelevant noises and provide a clean acoustic signal. Nevertheless, it would be interesting future work to use audio separation and denoising methods to separate the useful signal from the environmental noises.
> >
> > #### **4) It would have also been better to have a robot mounted camera to provide additional information for peg insertion:**
> >
> > A robot-mounted camera is one way to compensate for detailed geometrical information and eliminate the occlusion. To examine the effect, we perform a new set of experiments using a wrist-mounted RealSense D435 camera in addition to our original table-mounted camera.
> >
> > The average test success rate of this method is 75%, with the major failure mode occurring on the slanted bumps. Please refer to Table 1 of the additional supporting material for the detailed results. The main reason is that although the wrist-mounted camera can capture the visible geometrical information such as the location of the bump, it cannot characterize many other implicit yet non-trivial properties about the contact, such as the surface material as we show in our experiment. Therefore the wrist-mounted camera can provide part of the necessary information for the robot to complete the task, but not all of it. Please refer to the additional supporting material for more detailed implementations and discussions.
> >
> > #### **5) The tasks that were chosen to demonstrate the efficacy of the system were not really ideal to show the reason why you need all the different modalities. … better if you had done something like previous cargo transport like graham crackers? More concrete limitations.**
> >
> > The main intuition behind our choice of the tasks is that they always appear in our daily routines, and as humans complete these tasks with no effort, we seldom think about what our multisensory feedback has granted us. For example, in the dense packing task, we can see an object is somewhere in a box, but if we are grasping it then we can only know whether the object is stuck on other things if we are grasping it in hand. As mentioned in Sec. 2 of the main text, these tasks are still very challenging for the robot and many studies are trying to propose different solutions to these problems, and we are most inspired by the way that humans utilize their eyes, ears, and fingers to explore the environment and process the hidden clues embedded in these modalities. Furthermore, with the ablation study results and analysis presented in Sec. 4.2 and Sec. 4.3 of the main text, we show with our experiments that all the three modalities are important in these two tasks and the role they play.
> >
> > Nevertheless, thank you very much for the valuable suggestion about the cargo transport task, and this would be an interesting choice in our future work. Based on all your insightful feedback, we have updated the limitation section and included more discussions on limitations and potential future work.

---

> > > ### Comment · Reviewer_yYyH · 2022-08-25
> > > **Response to Author Rebuttal**
> > >
> > > Thank you for taking the time to do more ablation studies with the wrist mounted camera and force readings from the Franka. I will take another read through the paper and then update my review; however, I will most likely not complete this before the end of the rebuttal period as there are only 2 more days left.

---

### Official Review · Reviewer_YUYT · 2022-08-07

**Originality:** Fair
**Technical Quality:** Fair
**Clarity Of Presentation:** Very Good
**Impact:** 2

**Recommendation:**

Weak Accept: I recommend accepting the paper, but will not argue for my recommendation if the majority of other reviewers have a different opinion.

**Summary:**

This paper presents a multisensory robotic system that can leverage vision, tactile (vision-based sensor) as well as contact microphone data. They propose using a multisensory self-attention model to simultaneously leverage each of the sensory modalities and present results on a dense packing task, and a pouring task with pouring a fixed quantity of beans into a cup.

**Issues:**

- More details and analysis of failure modes on the real-world setting for the dense packing experiment
- Results on the pouring task would be interesting to see in a more general setting. Can multimodal inputs actually help the policy have a better sense of what quantity of material has been poured? It is unclear if this is a well-defined task because even humans are not very good at estimating quantity poured open loop without looking at the level to which the cup has been filled
- It is unclear how much visual input any of these policies need to be successfully executed. It would be useful to see a tactile+audio baseline on these tasks using a basic heuristic visual policy if required.

**Quality Of The Limitations Section:**

Limitations are addressed clearly

**Reviewer Expertise:**

4: The reviewer is confident but not absolutely certain that the evaluation is correct

**Robotics Focus:**

Sufficient demonstration on hardware

**Strengths And Weaknesses:**

Strengths:
- Proposes a simple attention-based architecture capable of combining visual, tactile and audio inputs.
- Demonstrates multimodal learning from just ten human demonstrations for each task.

Weaknesses:
- Dense packing experiment in the real world is lacking significant details on how the experiment was performed. What were the types of objects used? Were there harder and easier objects? How was success defined?
- Pouring experiment is way too specific and could just be overfitting. Even if you fix the number of beans you pour into the cup, at least vary the quantity of beans you start with.
- There is no real justification for using vision for these tasks and no baselines that use just audio and tactile sensing and a basic heuristic visual policy. The visual aspect of these tasks seems relatively trivial and achievable without needing to be learned.

**Summary Of Recommendation:**

While the results demonstrated in the paper are promising, the tasks are very limited in scope to convincingly demonstrate the effectiveness of this method for more general tasks. A number of details are missing on the dense packing real-world experiment, while the pouring task operates with fixed initial and and final conditions.

-------

I thank the authors for their detailed response and the extra details and experiments provided. While I still believe the tasks are limited in scope and the method involves minimal novelty in terms of technical contribution, it is a commendable engineering effort and shows a promising approach for learning from multiple modalities.

---

> ### Author Response · Authors · 2022-08-24
> **Response to Reviewer YUYT**
>
> Thank you for the constructive feedback!  We fully agree with you on the importance of the experimental setups and ablations you suggested; meanwhile, we would like to clarify that exactly because we are on the same page, we have indeed provided these details and results in the original submission. Please see our detailed responses below. We understand that any misunderstanding might be a presentation issue, and have also revised our submission to make the presentation clearer.
>
> #### **1) Details of real-world dense packing experiment:**
>
> We would like to clarify that we have discussed the implementation details of our real-world dense packing experiments in the original submission, in particular in Sec. 3.3 of the main text and Sec. 4.1 of the supplementary material. The goal of this task is to fit a glass being grasped by the robot into a crowded box, and the glass must reach the bottom of the box without destroying other objects around it. The box is densely packed with various objects and there is only one hollow space left in the middle, as shown in Fig. 1 of the main text. With the camera, the robot is able to locate where the empty spot is, yet there are small objects arranged in the bottom of the box which cannot be visually observed. To complete the task, the robot must correctly identify the geometry and material of the object through vision, audio and touch.
>
> Compared to the packing task where we use the 3D printed peg and bases, we use more diverse real-world objects in the real-world dense packing experiment. Around the glass, we have a wooden box, a thin metal board, a plastic plate, and harder deformable objects such as a soft stuffed toy and a towel. When the robot tries to complete the task, the glass will interact with these objects, and it must respond differently.
>
> #### **2) Pouring experiments could just be overfitting, … vary the quantity of beans you start with:**
>
> This is a great point, but we wish to clarify that we actually **are** using various initial conditions already, as shown in Table 3 in the main text, and the trained models **do generalize** to unseen initial weights of the beans, as shown in Table 1 of the supplementary material. These results have shown that the pouring experiments are **not** overfitting. Please see Sec. 3.3 of the main text for a more detailed description of our task setup.
>
> Specifically, from these experiments, we observe that the initial weight of the beans could have a large impact on the result if the model overfits to a fixed policy. The amount of beans affects the location of the center of mass of the cup; therefore, even if the robot follows one exact same trajectory, the rotation pattern of the cup will slightly differ depending on the weight of the beans and cause various pouring speeds. Without touch, the robot cannot interpret the pose of the grasped cup, so it always pours too fast and misses the correct timing to stop pouring. Please see Sec. 4.3 of the main text and Sec. 2 of the supplementary material for a more detailed discussion and analysis about the experiment results.
>
> In addition to varying initial conditions,  we have also tested the model’s generalization ability by shifting the location of the fixed cup during testing, as illustrated in Fig. 4b in the supplementary material. We have observed that if the models overfit to certain scenarios, they will fail to capture the cup locations through vision, which causes the robot to pour many beans out of the fixed cup.

---

> > ### Author Response · Authors · 2022-08-24
> > **Response to Reviewer YUYT (continued)**
> >
> > #### **3) Justification of using vision for these tasks and useful to see a baselines that use just audio and tactile sensing:**
> >
> > We agree with you on the importance of such justifications. Please note that in Fig. 5 of the main text, we have illustrated how the attention scores of each modality change during the tasks. From these demonstrations, we show that for the packing task, vision is most important for the robot to locate the box and when meaningful acoustic and tactiles signals are no longer available at the end of the insertion process. On the pouring task, vision is the most critical modality in that the two cups must keep aligned throughout the pouring process. As we see in Fig. 5b, the attention score is high on vision both at the initial aligning phase as well as during pouring, this is because the robot needs to align the cups as it keeps rotating the in-hand cup. Without a learned visual encoder that can adapt to the entire process, it would be very difficult for the robot to accurately pour the beans into the fixed cup. Please see Sec. 4.2 and Sec. 4.3 of the main text and Sec. 3.1 of the supplementary material. for more detailed discussions about how the robot effectively retrieves and fuses the information from each modality at different stages of the tasks.
> >
> > To further justify the necessity of the visual modality, per request, we have performed a set of additional experiments on the dense packing task that only use audio and touch without any learning on vision. When no visual input is provided to the robot, the robot always fails the task because it cannot move to the base with the initial position being random. To compensate for this problem, we adopt a heuristic visual policy as suggested during the initial aligning phase so that the robot can successfully reach somewhere close to the base from a random initial position. We apply the same visual heuristic policy during training and testing, and have the model learn over only audio and touch. The resulting average success rate is 0.80 over the four bases compared to 1.0 if learning over all three modalities. The major failure mode occurs on the hard slanted bump, where the robot successfully avoids the bump but fails to continue inserting downward after the signal disappears in audio and touch. At this time, noises may occur in the acoustic and tactile signals since the peg could still make contact with the inner walls of the base, so vision becomes more important to the prediction. Given that the final peg position differs on each base, a visual policy that is learned together with the other two modalities is non-trivial at this moment to inform the robot that the task is not yet completed. We have included this additional baseline in Table 1 of the revised submission.
> >
> > #### **4) Analysis of failure modes on the real-world setting for dense packing:**
> >
> > During the real-world dense packing task, the major failure modes we observe are as follows:
> >
> > 1) As shown in our supplementary video, one most common failure mode is that the robot cannot recognize whether the obstacle is hard or soft, thus pushing too hard onto the surface and getting stuck. When touching hard surfaces, such as the wooden box, the metal plate, or the plastic plate, the robot should move to its left or right to avoid breaking them. If the object is soft, such as the toy or the towel, we can just squeeze down because they can deform and give more space. However, in some baseline models, the robot does not successfully recognize the plastic plate and tries pushing the glass down, which is not allowed as it would be dangerous in the real world.
> >
> > 2) Another failure mode is that when we test some baseline models, the robot misses the correct range of the empty space in the box during the initial aligning phase, and the glass is moved out of the box. The main reason behind this is that the model does not retrieve the correct information from vision and attend to it, therefore the action prediction is confused by the inputs from the other two sensory inputs.
> >
> > As shown in Table 2 of the main text, our MULSA model provides the most stable and highest success rate among the three studied models and has the lowest chance of failing on the common issues discussed above. We have revised the text of Sec. 4.2 of the main paper to include these discussions.

---

> > > ### Author Response · Authors · 2022-08-24
> > > **Response to Reviewer YUYT (continued)**
> > >
> > > #### **5) Can multimodal inputs help estimate the quantity in pouring because even humans are not very good at it without vision?**
> > >
> > > Pouring is an interesting and challenging task, and rich information is embedded in different modalities during this process. Although in some scenarios humans can simply watch the water level, we are most inspired by how other hidden clues are interpreted and utilized. For example, we can tell whether a cup is empty, half-full, or almost full by listening to the sound of water being poured into the cup, and when we pour water out from a cup we can roughly estimate the amount of water left based on the weight change.
> > >
> > > Some previous works focused on the pouring task have proposed interesting ideas, such as ``Making sense of audio vibration for liquid height estimation in robotic pouring” (Liang et al., 2019), where the authors train a model to estimate the water height with sound only. In our experiments, we compare the performance of the model that uses all three modalities with the models that only use part of them. The results show that the model with full access to three modalities outperforms the others as shown in Table 3 in the main text. From the experiments, we see that audio is useful in deciding when to stop pouring, and touch is important for the robot to control how fast to pour. These findings suggest that with the help of multiple modalities, the robot can pour a more precise amount of beans more stably. We have also provided examples of the experiments in the supplementary video (11:01).
> > >
> > > We sincerely hope that our responses and changes have addressed your concerns and clarified any misunderstandings so that you would become more positive about the submission.  Please don’t hesitate to let us know if any concerns or questions remain, and we would love to discuss more.  We would like to thank you again for your time and review!

---

### Author Response · Authors · 2022-08-24
**Revised Main Paper and Supplementary Material**

Revised Main Paper and Supplementary Material

---

### Meta-Review · Area_Chair_GVJR · 2022-08-12

**Recommendation:** Accept (Poster)
**Confidence:** 5

**Metareview:**

The paper presents a multisensory attention model that can extract relevant features from the different modalities (RGB-D, visuo-tactile, and microphone sensors) for downstream manipulation tasks.

The paper **strengths** are:
- The use of multiple sensors in a single and simple pipeline for downstream tasks.
-The shown efficiency of the methods, in terms of needed demonstrations.
- The engineering effort to set up and provide real-hardware experiments.

The paper **weaknesses** are:
- Lack of comparisons and discussion of other methods that would use more than one modality, in tasks like pouring.
-Clear description of the experimental setup, and additional variations of the experiments to explain generalization of the proposed method.
-Limitations in convincing ablations stating from visual only data, to assess the significance of the added sensors on the various tasks.
-The method's limitations are not well addressed.

The authors should provide convincing answers to the reviewers' questions, and address the weaknesses of their paper.

**Post-rebuttal assessment:** After carefully inspecting the rebuttal discussion and the updated results, I appreciate the presented experiments and the addition of a new ablation study. Methodologically, I would argue that the use of Attention in Multisensory Fusion has been demonstrated previously (e.g., https://openreview.net/pdf?id=nhnJ3oo6AB), but not for the modalities and tasks presented in this paper, therefore making it an interesting application paper for the community.

**Best Paper Nomination:**

No

---

> ### Author Response · Authors · 2022-08-24
> **Revised main paper and supplementary material**
>
> Thank you for all the valuable feedback. We are glad to see that the reviewers like our idea of leveraging multisensory data for robotic manipulation tasks, and the promise of our robot system and experiment results. There are mainly some questions about some task design details and additional baselines, which we address in our separate responses below accordingly. We have also uploaded the updated paper and supplementary material to reflect the changes (in blue) that we make.
>
> Our main changes and clarifications are as follows:
>
> * We have clarified that we have indeed provided some requested details and experiment results in the original submission. We have updated our presentation to avoid any misunderstanding of the generalization experiments.
> * We have updated the main text and the supplementary material to include more detailed descriptions and discussions of our experimental setup, failure modes, and limitations.
> * Based on the reviewers’ suggestions, we have included a new ablation study on using the Franka robot’s force sensing, a new ablation study on using the wrist-mounted camera, a new ablation study to analyze the roles of magnitude vs. frequency in the acoustic signal, and a new baseline that uses audio and touch with a visual heuristic policy.